# Anti-Inflammatory Effect of Vitamin C during the Postoperative Period in Patients Subjected to Total Knee Arthroplasty: A Randomized Controlled Trial

**DOI:** 10.3390/jpm13091299

**Published:** 2023-08-25

**Authors:** Ricardo Ramón, Esteban Holguín, José Daniel Chiriboga, Newton Rubio, Carlos Ballesteros, Marco Ezechieli

**Affiliations:** 1St. Vincenz Hospital Location Salzkotten, 33154 Salzkotten, Germany; m.ezechieli@sjks.de; 2QRA, 170184 Quito, Ecuador; esteban@holguin.ec (E.H.); dr.josedanielchiriboga@gmail.com (J.D.C.); carballesteros@yahoo.com (C.B.); 3Ministry of Public Health (MSP), 170702 Quito, Ecuador

**Keywords:** Vitamin C, total knee arthroplasty, CRP, ESR, pain, opioids

## Abstract

Vitamin C, a potent reducing and antioxidant agent, plays an important role in the body, aiding in the growth of cartilage and bones. It is also involved in mechanisms that help reduce inflammation and its effects on the body. In addition, vitamin C decreases pro-inflammatory cytokines, such as IL-6, which produce acute-phase proteins such as CRP and influence inflammatory markers such as ESR. We carried out a study with 110 patients who underwent total knee replacement surgery. We divided the patients into two groups, in which the intervention group received 15 g of parenteral vitamin C during the immediate postoperative period while the control group did not. Patients who received 15 g of vitamin C after total knee replacement surgery had decreased inflammatory markers, specifically CRP and ESR. Overall, administering vitamin C in the post-surgical period results in improved management of inflammation, as evidenced by a decrease in CRP and ESR values. This leads to faster recovery and better healing outcomes for patients undergoing total knee replacement surgery. Furthermore, the beneficial effects of vitamin C in reducing proinflammatory cytokines, reducing the need for opioid analgesics, and its mild adverse effects make it a promising adjuvant in managing postoperative recovery.

## 1. Introduction

Vitamin C, also known as ascorbic acid, is a potent reducing agent and antioxidant that plays a crucial role in various metabolic processes [1]. It acts as a cofactor for enzymes such as Proline Hydroxylase and Lysine Hydroxylase, which are essential for the growth and strength of cartilage, bone, and subcutaneous tissue fibers [2,3]. The deficiency or absence of vitamin C can lead to defective collagen fibers, scurvy, and prolonged wound healing [4,5].

Vitamin C exhibits unique behaviors in the body compared with other low-molecular-weight substances, as its absorption, distribution, metabolism, and excretion processes are remarkably complex. Given its hydrophilic nature, it relies on cotransporters to facilitate its transport, creating significant concentration gradients. These cotransporters belong to the sodium-dependent vitamin C transporter (SVCT) family, which transports both sodium and ascorbate across membranes. SVCTs are present in all organs of the human body and exhibit a low-capacity/high-affinity relationship with vitamin C [3,6,7]. As a result, vitamin C distribution and organ concentration are highly compartmentalized, with some organs having low concentrations, like muscles, and others having large concentrations, such as the brain [3,6,7].

On the other hand, ascorbic acid is known for its antioxidant properties, which can neutralize free radicals through different mechanisms. These mechanisms include the direct elimination of free radicals and reactive oxygen or nitrogen species (ROS/RNS), the down-regulation of ROS/RNS-producing enzymes, the facilitation of other cellular antioxidants, and the activation of the signal pathway for NrF2 (Nuclear Factor Erythroid 2-Related Factor). The significance of Nrf2 in resistance to oxidative stress has been demonstrated by studies on Nrf2 knockout mice, which exhibited heightened susceptibility to chemical toxicity and diseases related to oxidative pathology [8,9,10]. According to Li and Zhu [8], vitamin C’s anti-inflammatory effects stem from its ability to inhibit the production of reactive oxygen species (ROS) and reactive nitrogen species (RNS) while simultaneously activating Nrf2 and dampening the activity of NF-B, all of which help to shield the body from the damaging effects of inflammatory processes. Within the human body, vitamin C works to diminish oxidative stress and inflammation [11,12]. Furthermore, the administration of intravenous ascorbic acid in substantial doses has been observed to have no discernible deleterious consequences [11,12].

Vitamin C is an essential component that helps boost immune defense by holding properties that make it an antioxidant and an anti-inflammatory agent. It achieves this result by enhancing the cellular activity of both the innate immune system and the adaptive immune system [13,14]. Increasing the efficiency with which immune cells function is another benefit of vitamin C. This immune-boosting quality allows for a more robust response to a wider range of dangers, including infections, trauma, and oxidative stress [13,14].

Ascorbic acid has been shown to provide analgesic effects, and large doses can reduce subjective pain symptoms and the need for painkillers. However, the precise mechanisms through which vitamin C provides this soothing effect have not been fully elucidated [9,15]. Additionally, a meta-analysis and systematic review performed by Aïm et al. [16] concluded that the use of vitamin C reduces the risk of developing a complex regional pain syndrome after orthopedic surgery. Furthermore, a growing body of evidence supports the hypothesis that the administration of vitamin C exhibits analgesic characteristics in specific clinical circumstances. It has exhibited a remarkable reduction in the requirement for opioid medications in the management of pain across various contexts, including but not limited to orthopedic diseases and surgeries, oncological conditions, and viral infections [12,17,18].

During surgery, the body responds to controlled injuries to tissues by inducing a state of inflammation where proinflammatory cytokines are synthesized, primarily interleukin 6 (IL-6), which is responsible for mediating the production of acute-phase reactants like C-Reactive Protein (CRP) [19,20,21]. These parameters can be measured to determine if a systemic inflammatory process is underway, along with other parameters like the Erythrocyte Sedimentation Rate (ESR). Fluctuations in CRP and ESR levels, among other inflammatory markers, have been quantified over time and have been found to reach their highest levels during the perioperative period. Subsequently, these levels gradually return to baseline within a timeframe of approximately 2 to 6 weeks [19,20,21].

Several previous studies have established a correlation between total knee arthroplasty (TKA) and the blood levels of acute phase reactants, indicating a significant association. In an effort to mitigate this pro-inflammatory response, various pharmacological interventions have been proposed with the aim of improving clinical outcomes and reducing adverse effects. One notable study by Nielsen and colleagues [22], which employed a double-blind randomized controlled design, demonstrated that patients undergoing TKA who received a high dose of dexamethasone exhibited a reduced inflammatory response, as evidenced by lower CRP levels at 24 and 48 h, as well as improved analgesia.

Taking into account the findings from the aforementioned literature, it appears biologically plausible to consider the administration of vitamin C to orthopedic patients undergoing TKA as a means of attenuating the influence of pro-inflammatory molecules in the body. Vitamin C is known for its ability to suppress the production of cytokines, which are key mediators of inflammation. Considering factors such as CRP and ESR levels, recovery time, and pain, it is reasonable to hypothesize that these patients may experience an accelerated recovery following surgery.

## 2. Materials and Methods

We conducted a randomized, single-blind, single-center, controlled trial with a series of 110 patients who underwent unilateral or bilateral TKA between October 2018 and July 2020 at the DaVinci Surgical Center in Quito, Ecuador. All surgical procedures were performed by the same surgeon.

Detailed information regarding the study was provided to each patient, allowing them to make an informed decision regarding their participation. After explaining the potential benefits of publishing patient case data, patients gave their permission for their information to be utilized. Participants were given the opportunity to withdraw from the study at any moment during their participation in it.

Patient selection for the study was based on their willingness to participate, regardless of age, gender, or the presence of any adjuvant conditions or medications. The primary total knee arthroplasty procedures were performed at the DaVinci Surgical Center. Patients who had a history of an allergy to ascorbic acid, daily intake of vitamin C, a preexisting renal illness, revision surgery, or previous knee operations on the involved knee were not eligible to participate in the trial.

A total of 110 patients who met the inclusion and exclusion criteria were included in the study. The surgeon’s high standards remained unchanged from patient to patient. During the surgical recovery phase, the patients were randomly divided into two groups to evaluate the effects of high-dose vitamin C administered intravenously. In order to ensure the preservation of randomization throughout the procedure, the selection of the infusion type was conducted in a random manner by an impartial third party, namely, a nurse who possessed no knowledge of the study. The administration of the respective entities was indeterminate in this scenario, and only upon careful examination of the series was it possible to ascertain their appropriate order. In this particular scenario, a greater number of individuals were administered placebos. During the initial postoperative period, which refers to the first four hours after surgery, patients received an infusion without knowing if it contained vitamin C. The intervention group consisted of 45 patients who were given a parenteral dosage of 15 g of vitamin C. This amount corresponds to a “very high dose” (>12 g/day), although it did not meet the threshold for a “megadose” (50–100 g/day) [23]. The number of patients who participated in the control group was 65, and they all received an infusion that was devoid of vitamin C. Both groups were given opioid-sparing anesthesia in accordance with the best practices protocol of the Faculty of Pain Medicine of the Royal College of Anaesthetists and the established protocols of the Anesthesiology Department of the DaVinci Surgical Center. These protocols included the administration of a peripheral nerve block that was guided by ultrasound.

On the eighth postoperative day, all patients attended scheduled follow-up appointments. During these appointments, blood tests were carried out, and the levels of the inflammatory biomarkers CRP and ESR were measured in both groups. Median values were calculated using the collected data, which provides a more representative measure less influenced by outliers and offers a realistic depiction of typical values compared with what is often portrayed in the media. Patients were informed of the outcomes specific to their cases as soon as the relevant data became available; however, there was no discussion of the overall findings of the research. Post-procedure, each patient continued to attend their regular follow-up visits for up to one year.

The standard preoperative workup at the DaVinci Surgical Center typically consists of a radiographic evaluation in addition to blood tests, medical exams, and anesthesia evaluations. Aspirin was used as an agent for the prevention of venous thromboembolism, and the regimen for postoperative care included a fast-track recovery technique. Additionally, no opioids were administered to the patients. Mobilization and exercises were started on the day of surgery under the direction of a physical therapist. Patients were either discharged on the day that they had their surgeries or the day after. Patients went through the same rehabilitation process. Surgical staples were typically removed between fourteen and sixteen days post-surgery.

Following the surgical intervention, patients received weekly follow-up care for the first month, followed by monthly care for the subsequent three months. Afterward, they received follow-up care every six months for the first year and then annually.

As a result of the nature of the information that was gathered for this investigation, the Mann–Whitney U test was selected to analyze the results. The Mann–Whitney U test was selected as the appropriate method of analysis, more suitable than a T-test, because it is a non-parametric alternative that can be used to assess the statistical significance of the differences that can be found between groups and because we wanted to perform distributional comparisons rather than mean comparisons. In this particular instance, in order to ascertain the presence of a statistically significant distinction between the two groups, it was necessary to compute the sum of ranks. The rank sum may be conceptualized as the summation of the “ranks” assigned to the data points inside a certain group, following their arrangement in ascending order. This sum indicates whether there are appreciable differences between the two groups’ distributions.

## 3. Results

There was a total of 113 knees investigated in 110 patients who had undergone TKA, with just 46 (41.8%) of the participants being male and 64 (58.2%) of them being female. Among the knees that were operated on, 59 (52.2%) were the right knee, 48 (42.5%) were the left knee, and 6 (5.3%) were bilateral. The population that was researched had an average age of 66 years (the age range was 35–90 years).

CRP levels were determined using blood samples taken from participants in both groups who took part in the study. In the control group, the median concentration was 42 mg/L, with a range of 20–79 mg/L; in the experimental group, the median concentration was 21 mg/L, with a range of 8–42 mg/L. Between the two groups, there was a statistically significant difference (*p* < 0.001) (Table 1) (Figure 1).

Similarly, ESR values were determined; the median for the control group was 21 mm/h (range: 12–42 mm/h), whereas the median for the experimental group was 11 mm/h (range: 5–21 mm/h). The control and experimental groups’ ESR values were found to be significantly different from one another (*p* < 0.001), as shown in Table 2 and Figure 2.

Among the patients who had surgery, there was not a single reported case of infection, and none of them experienced severe limitations in their movement or any other major complications. It was noticed that all the patients in the experimental group were able to have the sutures removed between the fourteenth- and sixteenth days following surgery. In particular, the removal of the staples was accomplished in seven of the cases before the fourteenth day because the wound had healed sufficiently. Compared with patients who were part of the control group, a total of eighteen patients had their staples removed after a period of sixteen days.

During the first month after surgery, a noticeable trend was observed in the control group, indicating that patients who received vitamin C experienced reduced pain and improved mobility. This observation may be attributed to the earlier decrease in acute-phase proteins and inflammatory markers, which was evident during the examination and assessment of patient mobility.

Regarding the patients’ use of analgesics, it was not possible to carry out optimal controls or follow-ups that were entirely adequate or accurate. Therefore, this study did not gather sufficient evidence to definitively establish that the analgesics used by the control group were likely higher than those used by the experimental group. However, it was documented that, in the immediate postoperative period, the patients who received vitamin C required fewer rescue analgesics in comparison with the group that did not receive it. Nevertheless, additional research is required to shed light on whether there is a direct relationship between these qualities, which were noted only in a subjective manner by our research group.

## 4. Discussion

The administration of 15 g of vitamin C after surgery has been shown to improve the inflammatory process. It is also important to consider that trauma and surgery are known to significantly reduce vitamin C concentrations. Shah et al. [24] reported in their study that the patient undergoing elective total knee arthroplasty had suboptimal vitamin C levels prior to surgery, and 90% continued to have suboptimal vitamin C levels two days after surgery. In addition, Handcox et al. [4] determined that, after a micronutrient measurement, 54.4% of orthopedic trauma patients undergoing surgical fixation of lower extremity fractures were vitamin C-deficient. Vitamin C administered intravenously as opposed to orally would generate maximal plasma concentrations approximately 25 times higher, which supports the intravenous administration of vitamin C to increase its concentration in the body, thereby enhancing its efficacy [4,25]. 

The ability of vitamin C to influence the downregulation of hepatic mRNA that encode various pro-inflammatory cytokines, such as IL-6, is one of its many beneficial properties [1,21]. Traber et al. [2] proposed an additional mechanism, which is the downregulation of Hypoxia-Inducible Factor 1, also known as HIF-1, which, among other things, alters the function of neutrophils. Vitamin C provides anti-inflammatory qualities that are advantageous in the setting of a patient who has had total knee replacement orthopedic surgery. These features are caused by the processes described above. Because immune system cells react to this kind of harm and cause the activation of the inflammatory cascade, the systemic inflammatory response that occurs during and after surgery is something that should be expected. This reaction is caused by the controlled trauma that the tissues are subjected to during surgery. In addition, it has been demonstrated that vitamin C lowers NF-κB levels, which are a transcription factor that is implicated in the upregulation of proinflammatory gene expression [21,25,26]. NF-κB is an acronym that stands for nuclear factor kappa-light-chain-enhancer of activated B cells. This can be attributed to the inhibition of NF-κB activation caused by reactive oxygen species and the inhibition of a kinase caused by the oxidized form of ascorbic acid, which also helps to explain the results of this study [8,26].

Vitamin C has been shown to suppress the acute-phase response and the generation of inflammatory cytokines, in particular, IL-6, according to a study that was conducted by Ellulu, M. S., et al. [1]. According to the findings of other studies, taking a supplement containing vitamins C and E reduces the systemic plasma response of IL-6 by approximately fifty percent, which, in turn, prevents the release of IL-6 from contracting human skeletal muscle [21].

Vitamin C supplementation has been linked to better functional outcomes, decreased postoperative pain, and a lower risk of developing complex regional pain syndrome (CRPS) after orthopedic procedures. In addition to this, it is necessary for a wide variety of biochemical processes that have an influence on bone and skin health.

Regarding the use of vitamin C for the prevention of CRPS in cases of trauma and orthopedic disorders, there is considerable debate. Numerous studies have examined the efficacy of this antioxidant supplementation, both after injuries to the upper and lower extremities and in orthopedic surgery. The American Academy of Orthopaedic Surgeons and the Royal College of Physicians in the United Kingdom have updated their guidelines and recommended vitamin C supplementation as a moderate-strength recommendation for distal radius fractures. Nonetheless, this recommendation has not been restricted to a specific set of fractures. A number of researchers have conducted studies in recent years to determine whether vitamin C is beneficial in the treatment of post-traumatic and planned surgery involving the upper and lower extremities [9,10,16,17,27].

There is another component to all this, which is multimodal anesthesia or opioid-sparing anesthesia. It has been recognized that reducing inflammation correlates positively with diminished pain during the postoperative period. Given the current situation in some developed countries, and most dramatically in the United States, regarding the opioid epidemic, this opioid-sparing approach is now widely encouraged. There are various objectives according to the best practices published by the Faculty of Pain Medicine of the Royal College of Anesthetists, some of which are as follows: “All healthcare professionals involved in perioperative care should collaborate to provide the highest standards of patient-centered care including opioid stewardship,” and “Opioids should be used judiciously by healthcare professionals. This means using opioids when necessary, but stopping opioids when they are no longer required” [28]. With these objectives in mind, it is also reasonable to search for a safe and much-needed alternative to current pain management techniques, which the use of perioperative high-dose IV vitamin C could be adjunctive to. It was also demonstrated that the combination of antioxidants and anesthetics administered to elderly patients undergoing unilateral TKA under total intravenous anesthesia and lumbar sciatic nerve block can produce the desired sedative effect while reducing the anesthetic dosage. This may lead to the development of a new method of clinically minimizing propofol dosage [29].

Opioids can be recommended to orthopedic patients for the management of pain; however, these medicines have a number of side effects that might harm patients, including constipation, nausea, vomiting, QTc prolongation, and urine retention [30]. Vitamin C, on the other hand, may be purchased for a cheaper price and has fewer negative side effects than opioids. A randomized controlled trial by Jeon et al. [31] revealed that high dosages of vitamin C are beneficial for managing pain in the early postoperative phase. Because of this, providing large doses of vitamin C in the period immediately following surgery has two beneficial effects: first, it reduces inflammation, and second, it improves pain control and allows for reduced reliance on opioids [15]. It has been demonstrated that the use of this vitamin reduces postoperative morphine and opioid consumption and the need for rescue analgesics in patients undergoing musculoskeletal surgery [18].

In addition, the application of corticosteroids to the periarticular injection intraoperatively during TKA lowers the acute phase response, which also provides considerable pain reduction [32]. Studies have concluded that there is a positive relationship between inflammation control, pain control, and functional outcome. Given that the use of corticoids cannot be standardized across all groups, this has certain limits. This necessitates the use of alternative biological pathways to reduce inflammation and enhance patient outcomes in total knee replacement surgery [22,32].

It has been established that there is a considerable frequency of both macronutrient and micronutrient deficiencies among orthopedic trauma patients. Prealbumin, vitamins C and D, and zinc deficiency are among the most common dietary deficiencies. As a result of this, wound problems may result from a lack of prealbumin and vitamin C [2,3]. Vitamin C deficiency results in insufficient or deficient collagen enzyme production, which can result in unfavorable changes to the normal scarring process [2,3,17].

This study observed a statistically significant decrease in CRP and ESR values in the group of patients who received 15 g of vitamin C intravenously in the immediate postoperative period compared with the control group, suggesting a decrease in the inflammatory response and, ideally, a shorter recovery time. This result can be attributed to vitamin C’s ability to regulate the production of proinflammatory cytokines, such as IL-6, and inhibit the activation of NF-κB. Additionally, vitamin C has been shown to be effective in controlling pain in the immediate postoperative period, reducing the need for opioids and their adverse effects. The benefits are not only theoretical or visible in a laboratory given that it is known that adjuvant therapies that reduce inflammation and pain are a major component of opioid-free anesthesia, which has proven to be highly beneficial in reducing unnecessary risk for patients.

The intravenous administration of vitamin C is already widely utilized, and medical professionals need to be aware that their patients may seek intravenous vitamin C in addition to the traditional treatments that are currently available.

## 5. Conclusions

After carrying out the research, we determined that a considerable reduction in CRP and ESR values can be achieved when 15 g of intravenous vitamin C is provided in the post-surgical period. This indicates a reduced inflammatory response and, ideally, an improved recovery time. This—along with the previously cited literature indicating that pharmacologic interventions that reduce the inflammatory response of the human body to the trauma of TKA surgery are positively associated with less pain and better outcomes—opens the way for the investigation of new adjuvant treatments containing safer components, such as vitamin C.

These results are attributable to vitamin C’s multiple beneficial properties, such as its ability to regulate several proinflammatory mechanisms and its anti-inflammatory effects, as well as pain control and the use of opioid-free anesthesia, thereby decreasing the need for opioid analgesics and their side effects.

Vitamin C’s relatively modest and well-tolerated side effects also lend credence to this theory. Positive findings for the use of vitamin C in individuals undergoing orthopedic surgery were observed in this study.

Our study shows some encouraging findings, but more work is needed to establish a causal link between vitamin C delivery and better outcomes in orthopedic surgery. Future research should concentrate on finding optimal dosages and administration techniques for vitamin C in order to make the most of its potential health advantages while limiting any adverse effects that it might have. Clinicians should be aware that IV vitamin C is frequently used, and patients may request it in addition to traditional treatments.

In conclusion, the findings suggest that giving patients undergoing total knee replacement surgery a very high dose of vitamin C during the immediate postoperative period helps inflammation management. Vitamin C’s positive effects in lowering proinflammatory cytokines lead to a reduction in the requirement for opiate analgesics, facilitating better healing and faster recovery, and mild adverse effects make it a promising adjuvant for managing postoperative recovery. Continued research efforts in this sector will provide a better understanding of the efficacy, safety, and appropriate application of vitamin C in orthopedic therapy, leading to improvements in patient care and outcomes.

## Figures and Tables

**Figure 1 jpm-13-01299-f001:**
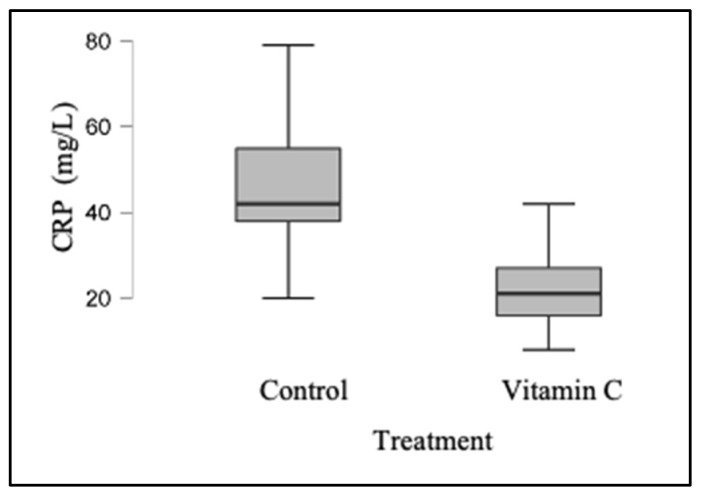
Graphic comparison of the distribution of CRP values obtained from both groups.

**Figure 2 jpm-13-01299-f002:**
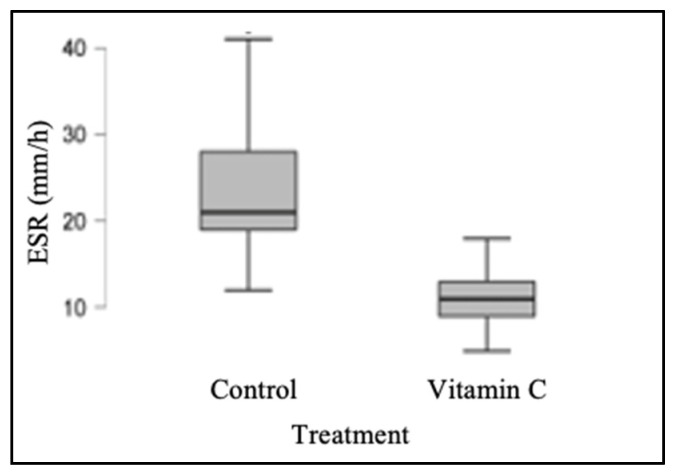
Graphic comparison of the distribution of ESR values obtained from both groups.

**Table 1 jpm-13-01299-t001:** CRP values for both the control group and the experimental group.

Group	CRP Value Median (Range), mg/L	Rank Sums
Control	42 (20–79)	2959
Vitamin C	21 (8–42)	984

Note: Mann–Whitney U test was applied; α = 0.05.

**Table 2 jpm-13-01299-t002:** ESR values for both the control group and the experimental group.

Group	ESR Values Median (Range), mm/h	Rank Sums
Control	21 (12–42)	1573
Vitamin C	11 (5–21)	510

Note: Mann–Whitney U test was applied; α = 0.05.

## Data Availability

Not applicable.

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
