# Peer review of "Anti-Inflammatory Effect of Vitamin C during the Postoperative Period in Patients Subjected to Total Knee Arthroplasty: A Randomized Controlled Trial"

_jpm, 2023, doi:10.3390/jpm13091299_

Round 1

Reviewer 1 Report

This manuscript describes and discusses the anti-inflammatory effect of vitamin C on the post-operative period in patients with total knee arthroplasty, using data evidence from a placebo-controlled trial conducted at DaVinci Surgical Center in Ecuador. The manuscript is well written in terms of language and presentation, clarify in terms of the trial design and trial conduct, and insightful discussions. I have some comments regarding the other aspects of this manuscript for the authors' consideration.

1. Trial design: this is a single-blinded, single-center, placebo-controlled trial. I have 4 observations regarding the trial design that the authors' explanations would be helpful: (1) how it was decided that out of the 110 eligble patients, 45 patients were given the vitamin C while the other 65 were given placebo. Why not plan 1:1 randomized ratio in this trial; (2) how patients were divided 2 groups? Was it randomzied based on patients' IDs? Was there any other factors considered in the randomization process? If there is another trial in the future, do the authors think a stratified randomization could be helpful? e.g., stratified based on right knee, left knee or bilateral; based on gender; (3) would the fact that the assignment was unblinded to the treating surgeon had an impact on the biomarker data and its interpretation? (4) Line 189-193 mentioned that a noticeable trend was observed in the control group, indicating that patients who received vitamin C experienced reduced pain and improved mobility. Given that it was one month after the surgery, and the patients already knew whether they took vitamin C or placebo at that point, would there be any impact of unblinding to patients on this conclusion?

2. Vitamin C usage during the trial: (1) vitamin C has its taste, how was the trial conducted to make sure on that specific day, patients did not know they took vitamin C or the placebo; (2) there can be other sources of vitamin C intake outisde of the trial in the daily life. So it is not clear what is the true differences in the cumulative vitamin C intake in the placebo group and the treatment group; (3) National Institute of Health (NIH) recommended daily dietary allowances for vitamin C to be 90 mg for male adult and 75 mg for female adult (https://ods.od.nih.gov/factsheets/VitaminC-HealthProfessional/). For the parenteral dosage of 15g that was used in this trial, could the authors comment on how much obsorption it is expected?

3. To show the anti-inflammatory impact, the authors used and evaluated the blood levels of two inflammatory biomarkers CRP and ESR. Could the authors discuss in the future, maybe clinically meaningful endpoints other than these 2 biomarkers could be used to evaluate the anti-inflammatory impact of a drug/treatment like vitamin C? It could either be clinical symptoms of inflammation based on the judgement of the treating physician, or functionality capacity of patients in the upper or lower extremity.

4. In Table 1, can the mean, SD, min, max, Q1 and Q3 also be reported. The se metrics will also help the understanding of the data, for example, the SD can help explain if these two groups are different in terms of the variations within the group.

5. Line 152 says as a result of the nature of the information used, Mann-Whitney test was selected to analyze the data. Could the authors explain further why using this test, and why not t test? Also, there can be other statistical analyses to be used, for example, ANCOVA model can be used to evaluate the impact of vitamin C on the biomarkers, adjusting for potential confounding factors such as gender or knee surgery categories. 

6. Ranks sum: it would be good if the authors could explain what ranks sum is, and how it was calcualted in the CRP value example and the ESR value example, as there will be readors who are not familiar with Mann-Whitney test.

7. The statement mentions that "Due to political issues in the country where the study was conducted in 2018, the protocol could not be officially approved by the local regulatory agency." As the protocol was not approved for this trial, is it ok to publish the data and the finding of the trial?

8. A very minor comment on the footnote of Table 1 and Table 2: maybe use "alpha=0.05" instead of "alpha=0,05".

9. Line 175 "Control an experimental groups' " maybe it is a small typo and should be "and".

Author Response

Dear Reviewer,

Thank you very much for the very accurate recommendations and comments, try to work as much as possible despite the short time given.

 Best regards,

Ricardo Ramón

Reviewer 2 Report

In this study, the researchers carried out a study with 110 patients who underwent total knee replacement surgery. Fort his purpose, the researchers divided the patient into two groups, in which the intervention group received 15 g of parenteral vitamin C during the immediate postoperative period while the control group did not. The researcher stated that patients who received 15 g of vitamin C after total knee replacement surgery had decreased inflammatory markers, specifically CRP and ESR. Overall, administering vitamin C in the post-surgical period results in improved management of inflammation, as evidenced by a decrease in CRP and ESR values. This leads to faster recovery and better healing outcomes for patients undergoing total knee replacement surgery. Furthermore, the beneficial effects of vitamin C on reducing proinflammatory cytokines, reducing the need for opioid analgesics and its mild adverse effects make it a promising adjuvant in managing postoperative recovery.

The researchers stated that this study observed a statistically significant decrease in CRP and ESR values in the group of patients who received 15 grams of vitamin C intravenously in the immediate postoperative period compared to the control group, suggesting a decrease in the inflammatory response and, ideally, a shorter recovery time.

As a result of the study, it was noted that this result can be attributed to vitamin C's ability to regulate the production of proinflammatory cytokines, such as IL-6, and inhibit the activation of NF-κB. Additionally, vitamin C has been shown to be effective in controlling pain in the immediate postoperative period, reducing the need for opioids and their adverse effects.

 Extensive source research has been carried out for the study. The methods applied in the study are described in detail. The findings obtained as a result of the study are presented in an understandable way with 2 tables and 2 figures.  The results are evaluated together with current sources. I believe that this study will provide important contributions to future studies on post-surgical period results in improved management of inflammation.

Author Response

Dear Reviewer,

Thank you very much for the comments.

 Best regards,

Ricardo Ramón